# Optimizing environmental enrichment for Sprague Dawley rats: Exemplary insights into the liver proteome

**Nathalie N. Roschke**[1], **Karl H. Hillebrandt**[1,2], **Dietrich Polenz**[1], **Oliver Klein**[3], **Joseph M. G. V. Gassner**[1,2], **Johann Pratschke**[1], **Felix Krenzien**[1,2], **Igor M. Sauer**[1]*, **Nathanael Raschzok**[1,2], **Simon Moosburner**[1,2]

1 Department of Surgery, Experimental Surgery, Charité –Universitätsmedizin Berlin, Corporate Member of Freie Universität Berlin and Humboldt-Universität zu Berlin, Berlin, Germany, 2 Berlin Institute of Health at Charité –Universitätsmedizin Berlin, BIH Academy, Clinician Scientist Program, Berlin, Germany, 3 Berlin Institute of Health, Center for Regenerative Therapies, Berlin, Germany

* igor.sauer@charite.de

**Data Availability Statement:** Proteomic data is available on the ProteomeXchange Consortium via the PRIDE partner repository (project accession no.: PXD046110). All other data is provided within

## Abstract

### Background

Considering the expected increase in the elderly population and the growing emphasis on aging-related biomedical research, the demand for aged laboratory animals has surged, challenging established husbandry practices. Our objective was to establish a cost-effective method for environmental enrichment, utilizing the liver as a representative organ to assess potential metabolic changes in response to differing enrichment levels.

### Methods

We conducted a six-month study involving 24 male Sprague Dawley rats, randomly assigned to four environmental enrichment groups. Two groups were housed in standard cages, while the others were placed in modified rabbit cages. Half of the groups received weekly playtime in an activity focused rat housing unit. We evaluated hormone levels, play-time behavior, and subjective handling experience. Additionally, liver tissue proteomic analysis was performed.

### Results

Initial corticosterone levels and those after 3 and 6 months showed no significant differences. Yet, testosterone levels were lower in the control group by the end of the study (p = 0.007). We observed 1871 distinct proteins in liver tissue, with 77% being common across groups. In gene ontology analysis, no specific pathways were overexpressed. In semiquantitative analysis, we observed differences in proteins associated in lipid metabolism such as Apolipoprotein A-I and Acyl-CoA 6-desaturase, which were lower in the control group (p = 0.024 and p = 0.009). Rats in the intervention groups with weekly playtime displayed the least amount of reported distress during inspection or upon room entry and were less prone to accepting treats. Removing animals from their enclosure was most effortless for those in

the paper and the supporting information files (S1 Table).

**Funding:** SM and DP received funding by the Charité – Universitätsmedizin Berlin, Charité 3R Research (no grant number available) (https://charite3r.charite.de). The funder had no role in the study design, data collection, analysis, decision to publish or preparation of the manuscript.

**Competing interests:** The authors have declared that no competing interests exist.

the large cage group. Over time, there was a decrease in conflicts among rats that interacted only twice weekly during playpen time.

## Discussion

In summary, refining husbandry practices for aging rats is both simple and budget-friendly, with no apparent adverse effects on stress levels, animal development, or relevant metabolic changes in the liver.

## Introduction

In the last years, refinement attempts towards improved laboratory animal husbandry on one side and a growing demand in aged laboratory rodents on the other, are challenging the established husbandry conditions and resources in research facilities. As the number of elderly people is expected to rise above 2 billion in 2050, biomedical research's increasing focus on aging and age-associated diseases equally requires more aged laboratory animals [1, 2]. In parallel, behavioral studies point to the assumption that the mere fact of keeping animals shielded from sensory stimuli, exploration, and activity opportunities is a constant stressor itself in common husbandry systems [3]. Refinement demands concerning handling and husbandry often exceed the readily available resources. While protocols and commercially available solutions are still lacking, scientists in Germany face concrete questions from authorities about strategies for refined housing conditions.

In animal-based research, refinement attempts to increase the "outer" methodological variation versus the need for standardized protocols for experimental research are often controversially discussed. The reported impact of differences in neurophysiology and behavioral sciences with respect to differences in housing conditions frequently limits further refinement. Therefore, scientists avoid increasing "outer" (protocol-part) variation assuming a loss of standardization, reliability, and reproducibility of research. Würbel and colleagues discuss the potential impact of "inner" variations developing from the standardized outer part methods that negate differences between individuals, strains, sex, and age [3, 4]. When individuals must cope with the very restricted standardized outer situation, the reaction (inner part), such as stress, will vary more depending on social hierarchy, age, and character and thus reduce reproducibility. In neurophysiology, an enriched environment led to effects including increased resistance to neurodegenerative processes, decreased anxiety, and shortened stress responses [5, 6]. This point of view prefers standardization via refinement-associated reduction of inner variation over outer standardization in husbandry.

As humans age, they become more susceptible to a wide range of internal acute and chronic diseases [1, 7–12]. Therefore, biomedical research increasingly focuses on age-associated conditions to meet the medical needs of elderly people. Based on former priorities, husbandries for laboratory animals mainly address breeding and short-term experimental protocols. This poses an interdisciplinary challenge for scientists and suppliers to proceed with adaptable variations of cage systems, enrichment tools, hygiene measures, and diets to serve the changing needs of aging animals. Particularly for rats, there are still few choices in commercial systems for the different needs of aging animals with relevant sex-related differences in body mass and social behavior [13].

Currently, minimal enrichment is given by a shelter, gnaw wood, and nesting material, but depending on the strain many mature males cannot even stand up straight on their hind paws

in the minimal required cages, tremendously reducing their quality of life [14]. Development and implementation of new housing systems, along with refinement demands and growing satisfactory experiences with animal experiments under controlled but higher "outer" variation, will need more time.

Meanwhile, we need fast and simple interim solutions for housing refinement. Furthermore, we need data-based information for scientists about what to expect concerning changed methods within the ARRIVE10-criteria, the comparability of former results with non-refined animals, and data to meet PREPARE criteria of new studies, such as required group sizes and potentially required changes in protocols and study design [15, 16].

As research on aging-related pathophysiology depends on expensive and time-consuming protocols (due to aging of the animals), it is crucial to collect and spread knowledge about the relationship between refinement-associated variation and the experimental results to achieve acceptance in the scientific community. The aging process of mammals involves multiple environment-organism interactions as well as physiological needs, and previous data on age-appropriate husbandry still leave many questions unanswered. We, therefore, aimed to share clarity for the following aspects:

- The impact of enriched husbandry conditions on the overall aging process and as one exemplary organ system–the liver.

- The minimum required financial resources based on choosing low-budget options for enriched husbandry conditions.

- The impact of the different enrichment programs on a standard marker for distress (corticosterone) since enrichment might increase social stress levels.

## Materials and methods

### Animals, group protocols and ageing

**Overview of the experimental design.**   All the projects were registered and approved by the local animal welfare office (Landesamt für Gesundheit und Soziales G0218/20). To determine the effect of different degrees of housing enrichment on the organ quality of rat livers, 24 male Sprague-Dawley rats were aged six months while exposed to enriched conditions and activity opportunities (Fig 1). Additionally, an evaluation of possible differences in behavior and development between the husbandry in larger groups compared to pair-housed individuals was analyzed. Upon arrival, all rats were pair-housed in standard cages for one week to acclimate to the facility before they were equally and randomly assigned to four groups consisting of six rats, respectively:

1. pair-housed in standard Makrolon type IV (**Control–C**)

2. pair-housed in standard Makrolon type IV with access to a playpen twice a week (**Control & Playpen–CP**)

3. housed in a group of six in a customized rabbit cage (**Large Cage–LC**)

4. housed in a group of six in a customized rabbit cage with additional access to a playpen twice a week (**Large Cage & Playpen–LCP**)

Two playpen variants were alternated to provide some diversion and stimulate explorational behavior. In addition, treats (sunflower seeds, oat flakes, hazelnuts, cashews, and almonds) were offered twice a week at playpen time for group CP and LCP. Inspecting and

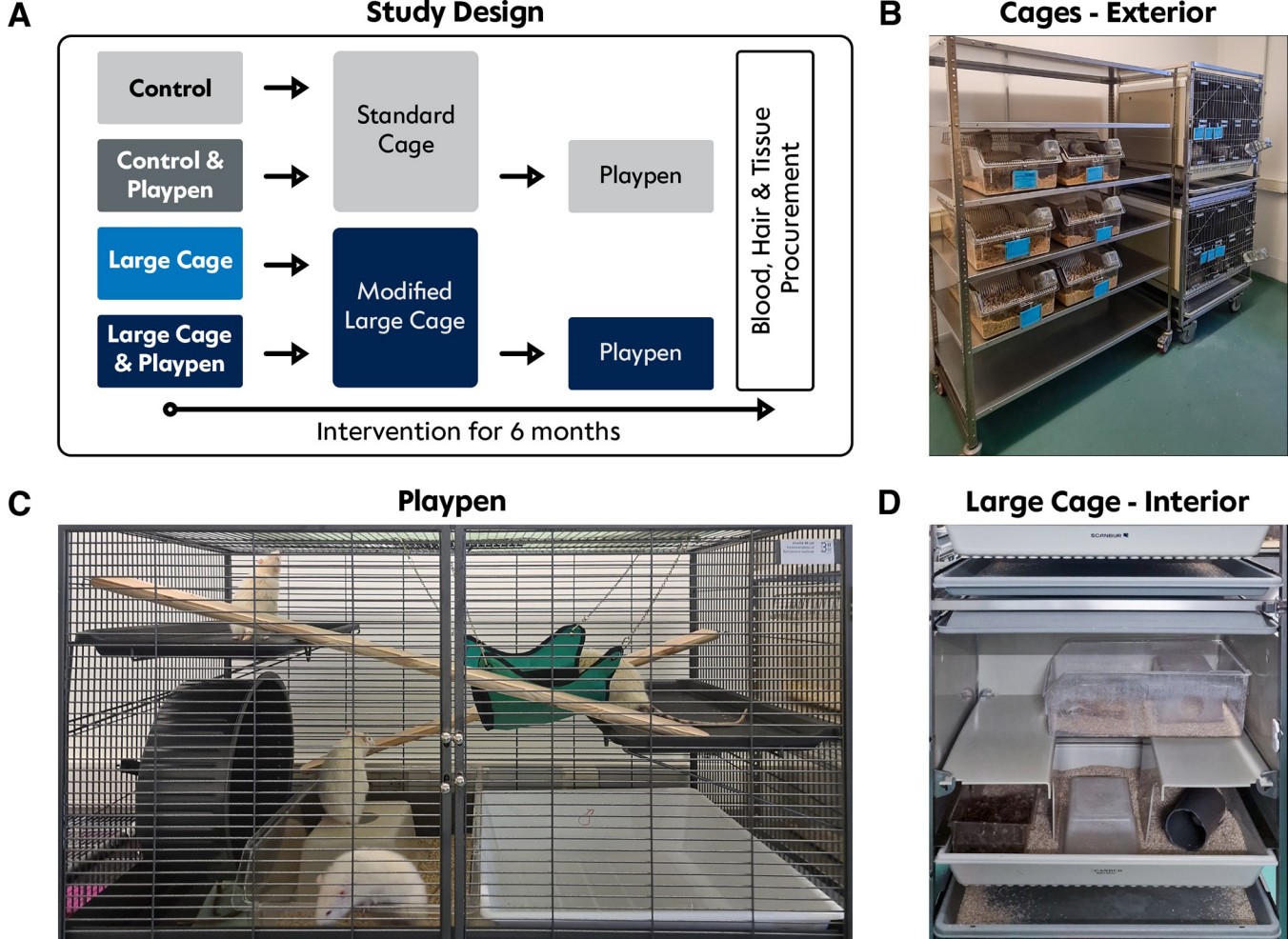

**Fig 1. Study setup. A** Study design of environmental enrichment. Rats were received at age 6 months and kept until 12 months of age in their respective group. **B** Exterior v of standard cages (left) and modified rabbit cage (right). **C** View of the playpen, which was available for two groups weekly. **D** Interior view of one of the larger cages. Images by Nathalie N. Roschke [2023].

weighing were performed to handle each rat at least once a week and to ensure good clinical health. Every month, hair samples were collected by shaving an approximately 1x1 cm-sized area at the lower back of the rats. Playpen times lasted at least 45 minutes, and up to 5 hours. During playpen time, the rats were video recorded for at least 20 minutes using a home CCTV camera (Shenzhen CTV Int Cloud Technology Co. LTD, YIMONA, approx. 15 fps). Footage of both light and intermittently dark periods was collected to investigate possible behavioral differences further.

**Animals.** This study used 24 male Sprague-Dawley rats, obtained by Janvier (Le Genest-Saint-Isle, France). At the beginning of the study, each rat weighed between 625 and 875 g (aged about six to eight months). Pathogen screening of the rat colonies is performed quarterly [17]. All procedures were conducted with approval from local authorities responsible for animal welfare and testing (Landesamt für Gesundheit und Soziales Berlin G0218/20).

**Caging.** The control cage unit was a standard Makrolon type IV cage (Tecniplast, Buguggiate (VA) Italy). This cage tray type measured 59.8x38x20 cm (length x width x height) and is used with a 10cm high cage lid (total height 30cm). It was used for groups (C) and (CP),

each composed of three pairs of rats. The customized cage used in this experiment was a former rabbit cage (Scanbur, Karlslunde, Denmark) which was reversibly modified for different levels to adapt to rat housing. It measured 80x60x60 cm (length x width x height). In addition, to guarantee the safety of rats of varying sizes due to different ages, the grid structure had been downsized to 1x1 cm to prevent animals passing through or getting stuck trying. Therefore, additional grids were fixated using screws to ensure they could be removed at any time. Gaps possibly broad enough for rats were reversibly occluded using plastic angle bars and cable ducts. The interior of the rabbit cages was constantly extended as a work-in-progress process. After about ten weeks, a transparent plexiglass was placed upon the standard rabbit houses, creating a second floor accessible through a central opening. Makrolon type III and IV cages were placed inside to further divide the space. Some of these cages were repurposed as burying boxes filled with standard bedding, sterilized coconut humus, paper towels, or various combinations (Fig 1).

**Acclimation and habituation.**  Animals were pair-housed for seven days in standard Makrolon type IV cages to allow acclimation to the facility before being randomly assigned to different housing groups. Another week of habituation was provided to groups LC & LCP previously to the first playpen time. The rats were marked individually per group by an increasing number of rings painted on their tails with black permanent marker. The marks were renewed every week. The rat was placed in the experimenter's lap for marking it, and the marking procedure was not started until the rat explored calmly.

**Husbandry.**  The rats were free-fed with pellet animal food (Sniff special diets–Rat/Mouse–Maintenance©, V-1534-000) and had access to municipal tap water via water bottles ad libitum. Additionally, treats were offered twice a week, varying between cashews, almonds, and hazelnuts combined with sunflower seeds or oat flakes. Every individual was hand-fed with one nut treat while about 10 g of seeds or flakes were scattered in the bedding to occupy the rats with seeking. Treats resemble about 4.1% of the overall food intake per month [18]. Cages were lined with hardwood bedding (LASbedding© Poplar Granulate 2–4 mm) and changed weekly. In addition, all four groups were provided basic enrichment consisting of opaque plastic tunnels, nesting material (paper towels, hay, straw), gnawing wood, and shelter made from used Makrolon type II cages that were semi-opaque. Beyond this, enrichment in the rabbit cages included the mentioned burying boxes and a running plate, both only inserted for a limited amount of time to keep it exciting and evaluate its impact on usage in playpens. Environmental influences such as temperature and relative humidity were held in the facility-typical ranges (20–27˚C; 15–80%). There was a 12h light and dark cycle with lights off at 6:30 p.m. Animals were housed under the controlled, group assigned, monitored, and documented conditions for 6 months.

**Playpen.**  Two alternating types of playpens were provided. All enrichment tools and items for activity were disinfected or autoclaved along with standard hygiene measures. The setup of the playpen interior evolved over time, reflecting the observed preferences of the animals. As Playpen version P1, a former feeding cart made of stainless-steel measuring 150x100x50 cm (length x width x height) was furnished with various items from within the facility: paper towel rolls, empty cardboard boxes, and sorted Makrolon type II and III cages. The latter were filled with either shallow water or the same burrowing materials as stated above. Nuts, Seeds, and flakes were hidden under paper towels and within the burrowing boxes. We provided climbing opportunities using grids, which eventually necessitated covering the trough with plexiglass plates to prevent escapes. Opaque open-ended tunnels, hay, straw, and the running plate were inserted intermittently to increase complexity. For occupation, a snack board toy (Trixie Snack Board) was placed at various times, filled with seeds or flakes. Playpen P2, in use until the 8th week, was a Savic® Nagerkäfig Zeno 2 cage designed

for home pet keeping. A Makrolon type IV cage was placed inside to provide familiar surroundings. Although it was furnished with the items mentioned earlier like Playpen version P1, its floor area offered less freedom of composition. From week 9 onwards, it was replaced by a larger cage, subsequently referred to as playpen version P2, which allowed for the addition of multiple ladders, tunnels, and two extra levels. The new playpen cage enabled the inclusion of a running wheel adequately sized for adult Sprague-Dawley males (380mm in diameter). We inserted animals into the playpens one at a time, which was found to be the most effective way to incorporate it into the process of weighing and marking the animals before moving them to the playpen. Therefore, the rabbit-caged group (6 animals) was just relocated, whereas the standard-housed group was only assembled during playpen times (3x2 males from their home cages) and subsequently separated.

## Handling

Each rat was handled at least once per week. Additionally, playpen groups were handled at least twice a week in the process of transferring them to the playpens and back to their home cages. The handling included the weighing and marking procedure by placing each rat in the experimenter's lap, as well as the weekly health check inspection. In the final phase of the study, blinded handling tests with beginners and advanced scientists, animal caretakers, and veterinarians were performed to investigate potential differences in handling procedure tolerance among groups. The test participants were asked to determine the body weight of four rats from each group followed by an evaluation of their behavior, such as the acceptance of treats or attempts to bite. Furthermore, the participants were instructed to evaluate the task's difficulty level in relation to challenges arising from potentially defensive or fearful behavior. The questionnaire included three to four statements depicting three specific aspects, alongside an overarching summary statement. The items used, covered the aspects of the rat's curiosity about interaction, the environment, and the behavior exhibited during the handling process. Subsequently, participants' agreement levels with the statements were used to assign scores, ranging from 1 (complete disagreement) to 5 (strong agreement) points. The selection of statements was based on the observable differences noted in preliminary tests and the experimenter's subjective viewpoint over the course of the study.

## Animal sacrifice

Under deep anesthesia achieved with inhaled isoflurane and subcutaneous buprenorphine and ketamine administration, as outlined in previous studies [19–21], animals were sacrificed and their livers were obtained. In short, after verifying the absence of pain perception, the rat's abdomen was sterilized and opened. Blood collection and liver flushing were facilitated by cannulating the abdominal aorta, and clamping the thoracic aorta. The liver was flushed with 20 mL of cold saline solution via both the aorta and the portal vein. The liver was subsequently fully isolated, removed, and tissue samples were snap-frozen or placed in 4% formalin.

## Blood and hair analysis

Hormone levels (testosterone, dehydroepiandrosterone, and corticosterone) were measured in hair samples, collected from the lower back by shaving, by Dresden LabService GmbH (Prof. Kirschbaum et al., Dresden, Germany). Hair samples were stored until the end of the intervention period after 6 months at room temperature in a dry, dark environment away from direct sunlight, and were collected monthly. Blood samples were biochemically analyzed for the pursued markers (ASAT, ALAT, AP, Urea, Creatinine, LDH, Albumin, and Glucose) by Labor Berlin–Charité Vivantes GmbH.

A blood gas analysis was conducted using the arterial blood drawn from the Aorta. Measurements were performed using ABL800 FLEX (Radiometer GmbH, Germany).

## Behavioral data from video content of playpen time

Playpen times were determined after pretesting with different individuals to align with natural activity in the rat's circadian rhythm and to provide a reliable timeframe for the observer over six months. Hence, playpen time was scheduled at the end of the day, starting between 16:00 and 20:00. Allowing for the recording of video footage during both light and dark phases. Starting in week two, playpen time was recorded for a minimum of 20 minutes, but frequently lasted more than 45 minutes. Unfortunately, four files were lost due to technical malfunction or data corruption. In analyzing the video data, several factors were considered: the number of fighting events, whether the running wheel or plate was used for actual running activities and if so, how often, and the usage of burrowing opportunities. Furthermore, an intelligence toy was regularly introduced in the playpens, and an assessment of how long it took for the group to empty it was performed. In principle, the playpen resembled an open-field activity situation during the light cycle. The activity, as evidenced by increased movement in the playpen, was greater during dark periods, even though no specific measurement was available.

## Assessment of liver tissue triglyceride content

Triglyceride determination in snap-frozen tissue sections was carried out using a standardized kit (Sigma-Aldrich, St. Louis, USA) containing free glycerol reagent and triglyceride reagent. The liver tissue's overall triglyceride content was measured after mechanical and thermal homogenization, followed by centrifugation and a 1:10 dilution, according to the manufacturer's provided instructions. Measurements of duplicate samples were taken at 540 nm using an Infinite 200 PRO plate reader (Tecan Group, Männedorf, Switzerland).

## Liver tissue proteomics

The rat liver samples were homogenized for 60 seconds at 25,000 rpm in an Eppendorf tube using the mixer mill MM400 (Retsch GmBH, Haan, Germany). After homogenization, the tissues were lyophilized and stored at -80˚C. The samples underwent protein extraction, digestion, and peptide desalting using the filter-aided sample preparation (FASP) method. For protein identification purposes, a NanoHPLC system (Dionex UltiMate 3000, Thermo Fisher Scientific, Waltham, MA, USA) connected to an ESI-QTOF ultra-high-resolution mass spectrometer (Impact II, Bruker Daltonic GmbH, 28359 Bremen, Germany) was used, loaded with 2 μL of the eluate. Proteomic data is available on the ProteomeXchange Consortium via the PRIDE partner repository (Project accession: PXD046110).

**Semiquantitative analysis of liver proteins.** The identification of peptides was conducted using PEAKS Studio (Bioinformatics Solutions Inc., Waterloo, Canada) with precise parameter settings, which included 20.0 ppm parent mass error tolerance and 0.05 Da fragment mass error tolerance. Monoisotopic precursor mass search type and trypsin enzyme were selected, with a maximum of 3 allowed missed cleavages. Up to three variable post-translational modifications (PTMs) per peptide were considered, including Oxidation (M), deamidation (NQ), and acetylation (N-term), taxonomy: Swiss-Prot database of Rattus norvegicus. Applying a peptide significance filter of -lgP >20 and a protein significance filter of -lgP >15, we set the unique-peptide filter to 1. The label-free quantification method was utilized, allowing automatic detection of the reference sample and alignment of the sample runs. We adjusted the protein significance filter to 0, the unique peptide filter to 1, and the protein fold change filter to 1. ANOVA analysis was then used to identify differentially abundant proteins. Based on the

rat genome database in the Multi Ontology Enrichment Tool (MOET) version 2.23, we performed Gene Ontology (GO) enrichment analysis.

## Statistical analysis

For statistical data analysis, R (version 4.3.0) and R Studio (version 2023.06.0) for macOS (R Foundation for Statistical Computing, Vienna, Austria) were used. Packages added for graph plotting and further analysis were *tidyverse* and *gtsummary*. Groups were compared using the Kruskal-Wallis test and repeated measures using the Friedman test. Bartlett test was used to test for homogeneity of variances. Spearman's rank correlation coefficient was used to compare correlations between hormones. Results were considered significantly different with a p-value < 0.05 and are reported as median with interquartile range (IQR).

## Results

### Designing a cost-effective enriched habitat

To maximize resource utilization, we started with a rabbit cage from the research facility. Our initial attempt, using chicken wire to close gaps, had safety and usability issues, and rats gnawed the cable ties. In the second approach, we used metal grids (kitchen supply) with smaller openings and acrylic panels to close gaps, allowing reversible customization. The interior design was adjusted over ten weeks using items like rabbit houses, Makrolon type III and IV cups, and short tubes from the animal care facility (Fig 1). Costs for modifying the double rabbit cage were about €400 (USD 421).

### Animal behavior and handling

All rats steadily gained weight throughout the analysis period. The median animal weight was 728 g (IQR 55 g) at baseline and 790 g (median, IQR 97 g) on the last day. The overall median weight gain between the starting point and the end point was 63 g (IQR 38 g; p > 0.9) (S1 Table). Rats showed individual preferences in using the different enrichment elements. The usage of vertical climbing possibilities was relatively low. Hiding in houses or tubes, however, was used extensively, especially in light periods. Regarding treat options, almonds and hazelnuts were observed to be preferred over cashew nuts. Rats living in the modified large cage were significantly less interested in the added running wheel or running plate during biweekly playpen time than group CP (Fig 2A and 2B). Over the study period, the running plate was used a median of 8 times (IQR 6, 21) in the CP group and only 4 times in the LCP group with the highest grade of enrichment (p<0.001). The same was observed for the running wheel (CP: 4, IQR 3.25, 6; LCP 0, IQR 0, p<0.001). The LCP group could solve the intelligence toy much quicker (5.6 minutes vs. 10 minutes, p = 0.029) and remained quicker over time (Fig 2C and 2D).

Handling of rats were rated by 10 participants, who took part in the questionnaire (5 beginners, 5 experienced in animal care and handling). Overall handling difficulty ratings differed among participants, based on their level of experience in working with rats. Inexperienced participants rated the procedure to be overall more demanding than experienced participants (0.9 to 1.2 rating points difference). However, in the LCP group, there was no difference between the level of experience. Overall, the ratings across groups showed no significant differences among intervention groups (Fig 3).

### Hormone analysis

Between start and end of the experiment, we observed corticosterone levels to first decrease from 16.98 pg/mg (IQR 4.34 pg/mg) to 12.66 pg/mg (IQR 3.37 pg/mg) and increased again to

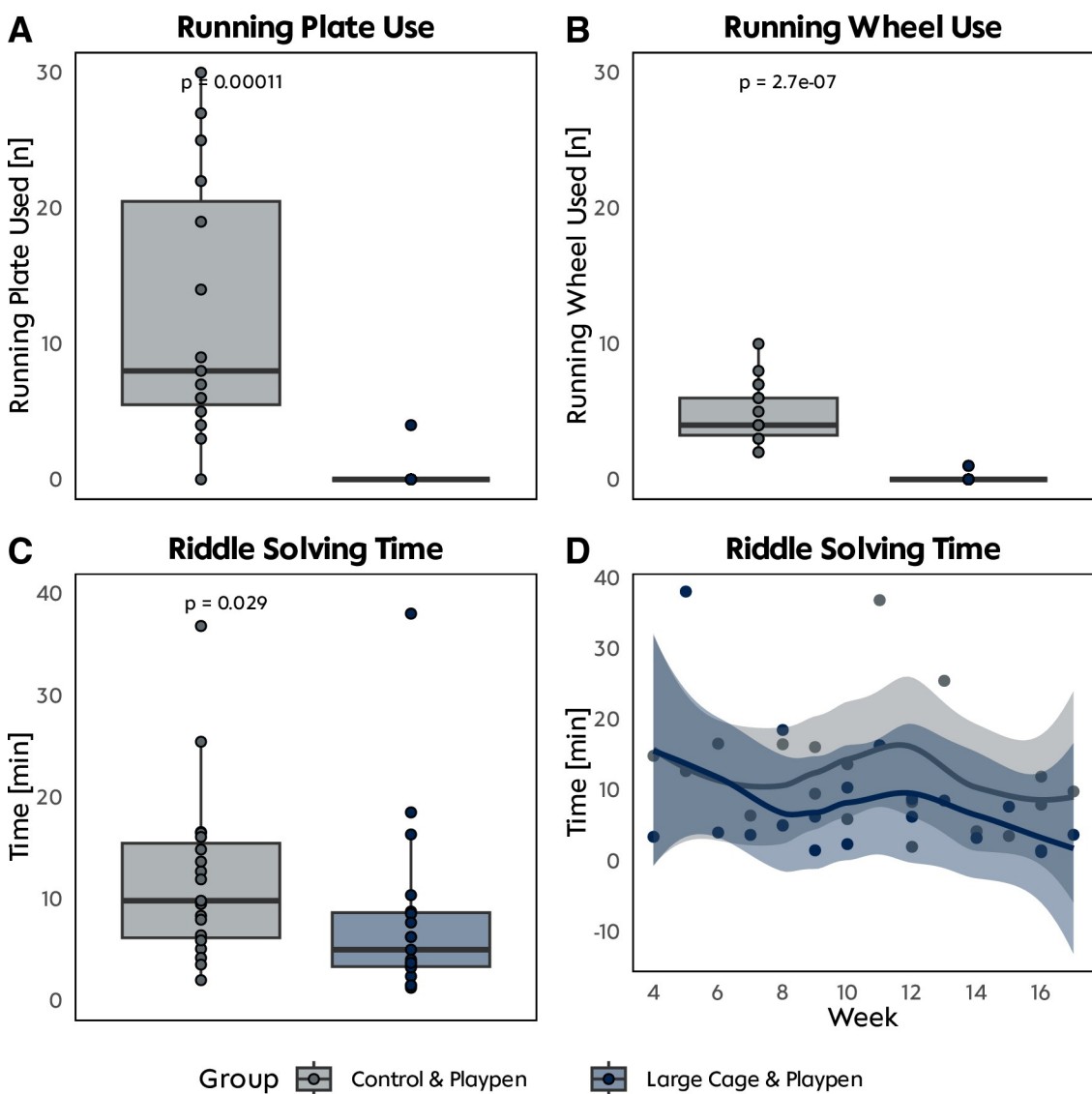

**Fig 2. Enrichment usage analysis. A&B** Number of times the running plate or running wheel added to the playpen were used by the individual enrichment group. **C&D** Time required to solve the provided riddle by group and over time.

16.8 pg/mg (IQR 4,6 pg/mg), reaching baseline levels (**Fig 4A**). No statistically significant differences between the group's median corticosterone levels could be observed at all three time-points (p = 0.48, 0.28, and 0.23), regardless of the level of enrichment. Moreover, a more detailed analysis revealed slight variations in the dynamics within the group's median hormone levels. Though not reaching statistical significance, the smallest reduction in the median corticosterone could be detected in group CP's values, whereas the control group (C) showed the most considerable decrease. Conversely, the biggest increase could be observed in the median corticosterone level of group LCP, which had the most refinement implementations.

Median testosterone levels slightly decreased from baseline values around 1.96 pg/mg (IQR 0.48 pg/mg) to a median of 1.51 pg/mg (IQR 0.47 pg/mg) (**Fig 4B**). Testosterone levels measured at the end of the experiment differed significantly between groups (p = 0.007), with the least amount of testosterone in the control group (1.08 pg/mg, IQR 0.22 pg/mg), 40% lower

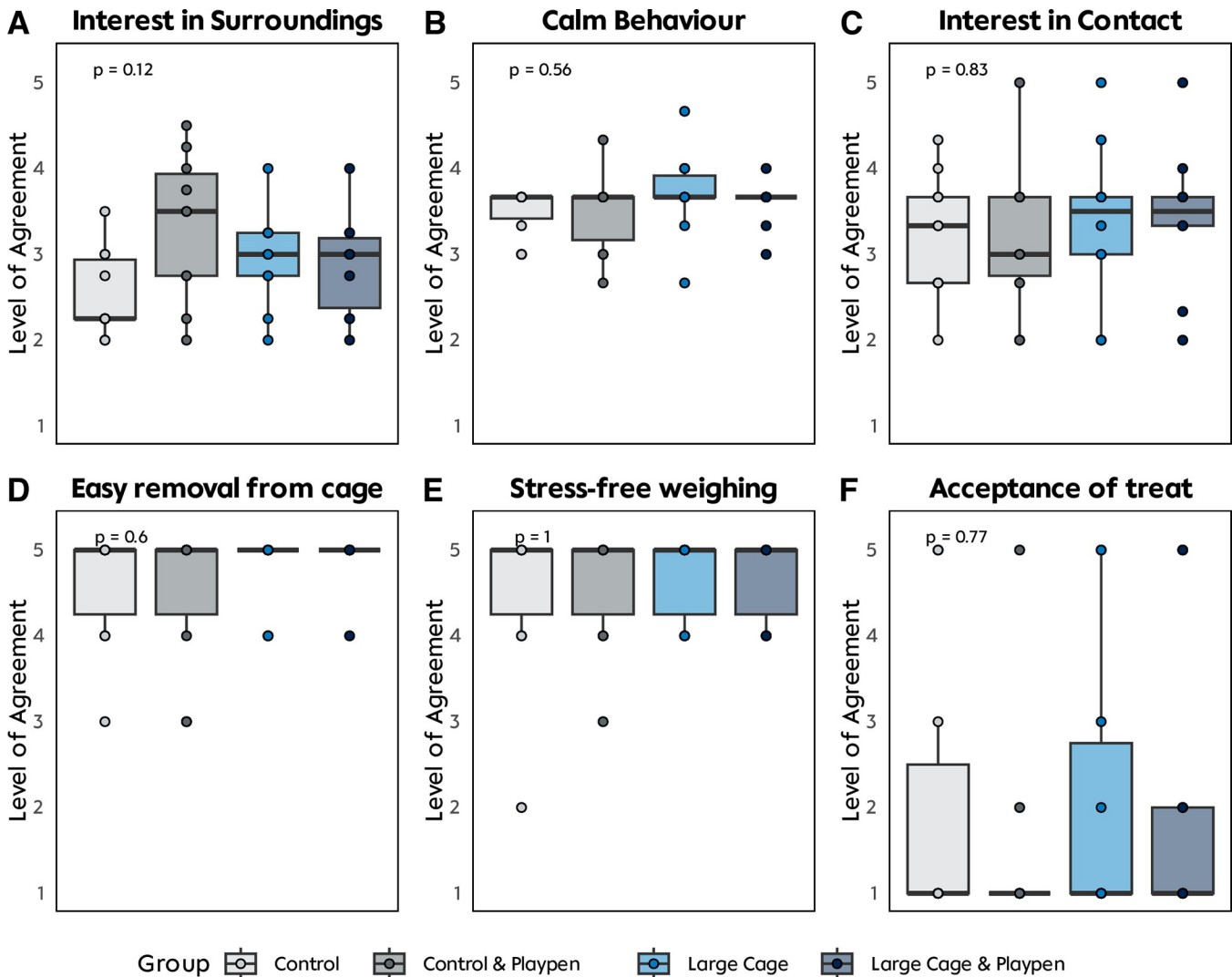

**Fig 3. Handling analysis. A-F** Results from the handling questionnaire performed after 6 months of intervention with 10 participants (5 beginner, 5 experienced in animal care and handling). Participants weighed and handled the animals and subjectively reported their experiences. No significant behavioral changes could be seen between groups.

value than the overall median. In the same manner as testosterone levels, the DHEA median decreased throughout the experiment, with significantly lower DHEA levels at the end point in the control group and the most refined group (p = 0.017) (**Fig 4C**). Surprisingly, the control group and the most enriched husbandry group (LCP) showed much lower variation within the group's DHEA levels (IQR 0.22 pg/mg, 1.05 pg/mg) compared to groups CP and LC (IQR 3.25 pg/mg and 3.14 pg/mg, respectively).

Overall corticosterone levels and testosterone levels correlated throughout the study period (**Fig 4D–4F**). However, when stratified by group at the start of the intervention, a correlation between testosterone and corticosterone levels was only found in group CP (spearman's R = 0.89, p = 0.018). A correlation between testosterone and corticosterone levels (R = 0.92, p = 0.009) was noted in group LC after three months. By the end of the study period, we observed a correlation between corticosterone and testosterone levels only in group LCP (spearman's R = 0.93, p = 0.007).

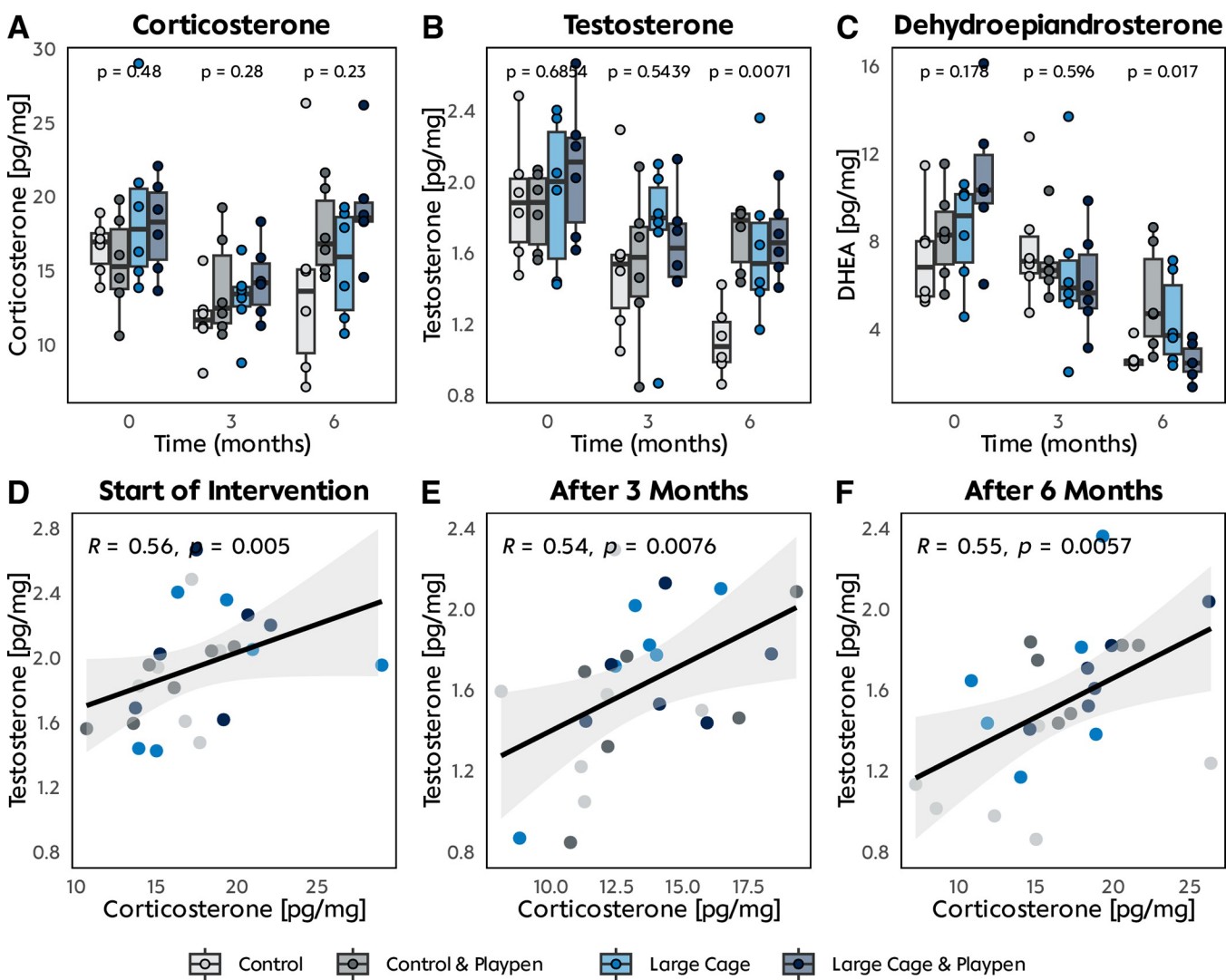

**Fig 4. Hormone analysis. A-C** Analysis of corticosterone, testosterone, dehydroepiandrosterone in hair collected at the start of intervention, 3 months, and 6 months. **D-F** Correlation analysis between levels of testosterone and corticosterone at the different analysis timepoints.

### Liver tissue analysis

**Biochemical markers.** Overall transaminase levels were within the physiological reference range (median AST 111 U/L IQR 29 U/L; ALT 104 U/L IQR 65 U/L) [22, 23]. While AST differences could not be considered statistically significant (p > 0.9), differences in ALT were remarkable between the groups (p = 0.022) (**Fig 5**). The lowest median ALT value was found for group (LCP), with the maximum enrichment, whereas the highest median value was observed for group (CP). Urea as a retention parameter was within the physiological range at an overall value of 6.16 mmol/L (IQR 0.58 mmol/L). Moreover, creatinine was considerably lower than the physiological reference with an overall value of 30.1 mmol/L (IQR 10.2 mmol/L) [22]. In the playpen groups, glucose level measures exhibited greater variation among individual animals compared to non-playpen groups (IQR$_C$ 0.79 and IQC$_{LC}$ 0.55 vs. IQR$_{CP}$ 2.89 and IQR$_{LCP}$ 2.59).

**Tissue triglyceride content.** Triglyceride quantification was performed to determine the impact of activity and especially nutritional enrichment on potential steatosis. Although the

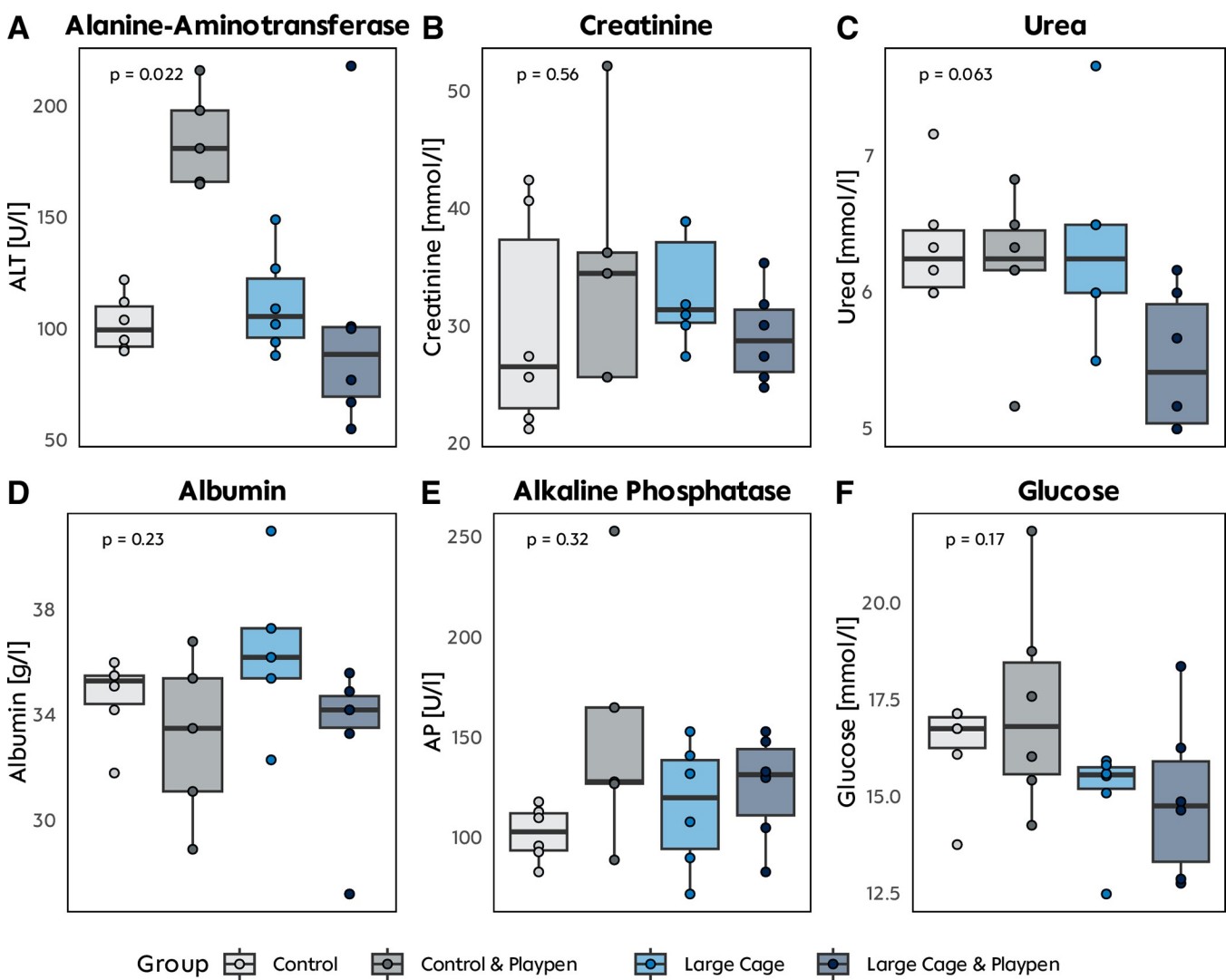

**Fig 5. Laboratory parameters. A-F** Biochemical analysis of blood drawn at the end of experiment. Only alanine-aminotransferase was significantly higher in the group with standard cage & playpen.

median total tissue triglyceride content did not differ between groups (overall median 26mg/dl, IQR 8mg/dl, p = 0.55), groups (LC, LCP), housed in the larger custom cage, tended to display higher values with 25 mg/dL (IQR 6 mg/dL) and 28 mg/dL (IQR 5 mg/dL), respectively.

**Proteomics.** We identified 1871 distinct proteins in our liver tissue samples. The majority (77%) was present in all different refinement groups (**Fig 6A**). Group LC had the largest proportion of distinct proteins (n = 69, 3.7%) and liver tissue from rats held in larger cages had more distinct proteins (n = 137, 7.3%) compared to rats held in Type IV cages (n = 86, 4.6%). Interesting proteins were ARMT1 in the LC group, a damage-control phosphatase in hexose phosphate metabolism, as well as squalene monooxygenase (ERG1) which catalyzes the stereospecific oxidation of squalene to (S)-2,3-epoxysqualene, and is a rate-limiting enzyme in steroid biosynthesis. Furthermore, the barrier-to-autointegration factor (BAF) was only present in the CP group, which is a non-specific DNA-binding protein that plays key roles in mitotic nuclear reassembly, chromatin organization, and DNA damage response. Finally, only the LCP group had PC4 and SFRS1-interacting protein (PSIP1), which play a protective role in

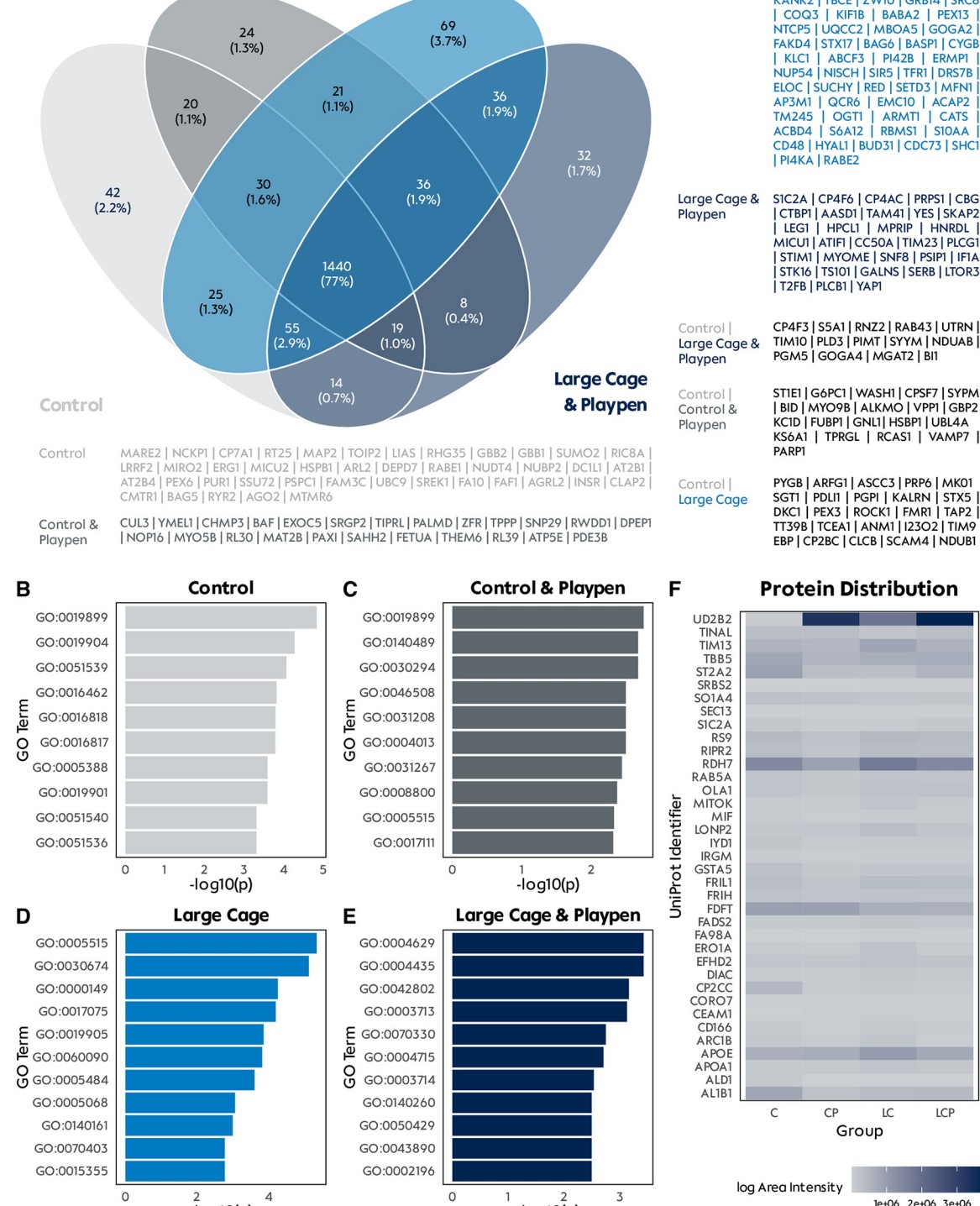

**Fig 6. Qualitative proteomics of liver tissue. A** Venn diagram of liver tissue proteins identified during proteomic analysis. Labeled per group are the distinct proteins found in each group by their UniProt identifier. **B-E** Gene ontology analysis of distinct proteins per group. Only the top 10 pathways are displayed and labeled according to their gene ontology term. **F** Heatmap of label-free quantification data of discovered proteins.

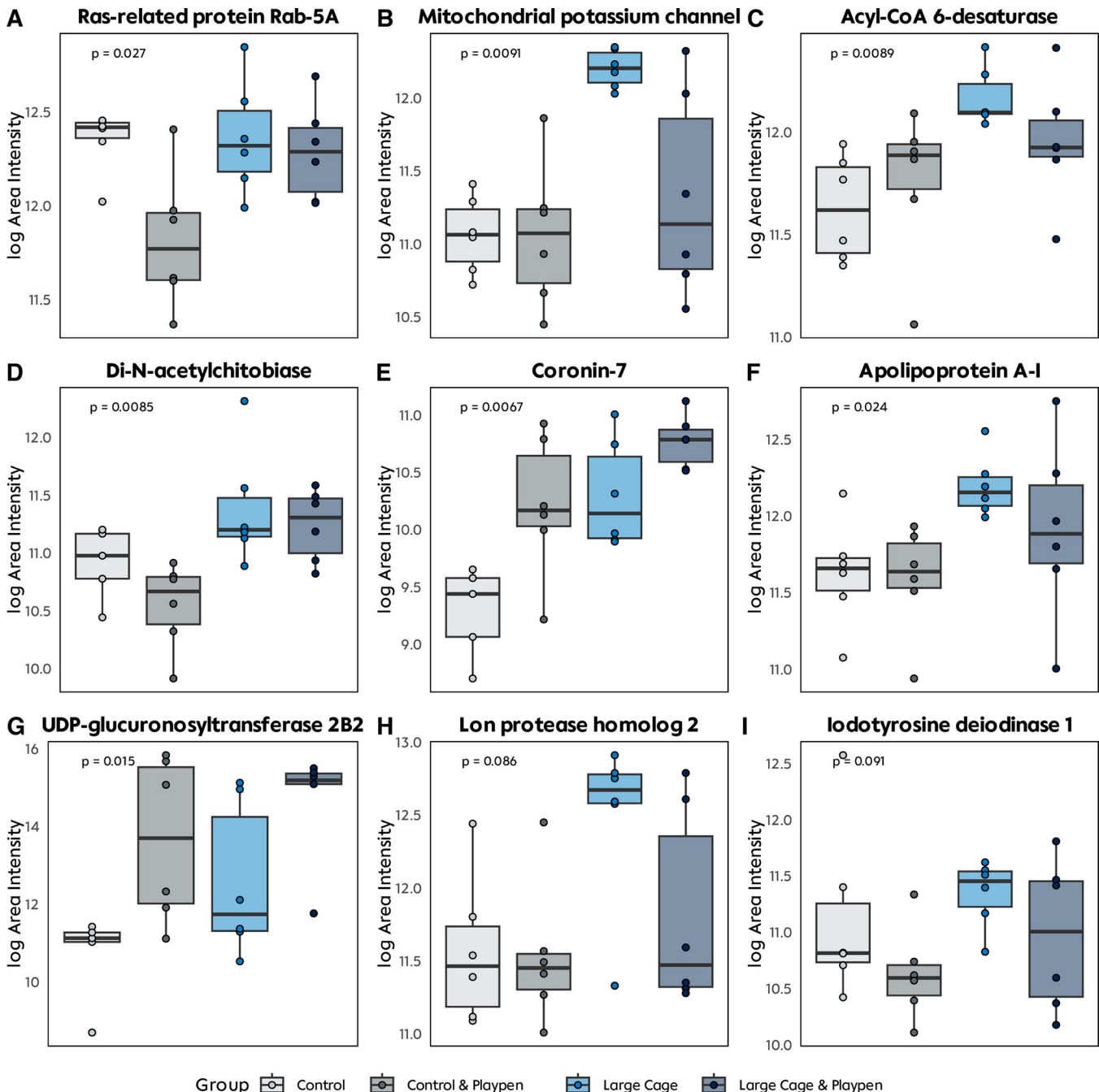

**Fig 7. Semiquantitative proteomics of liver tissue. A-I** Selected proteins from the label-free quantification dataset with significant or near significant differences in area intensity compared by group.

stress-induced apoptosis. However, gene ontology analysis of individual proteins per group did not reveal any overexpressed specific pathways (**Fig 6B–6E**).

In our label-free quantification analysis we saw several notable findings concerning the expression of specific proteins (**Fig 7**). Firstly, Ras-related protein Rab-5A demonstrated significantly lower expression levels exclusively in the CP group (p = 0.027). Conversely, the Mitochondrial potassium channel exhibited heightened expression in the LC group

(p = 0.009). Additionally, Acyl-CoA 6-desaturase, involved in the synthesis of fatty acids and regulation of steroid hormones, exhibited a remarkable trend of nearly linear relationship between its expression levels and the degree of refinement across all groups (p = 0.009). Furthermore, Di-N-acetylchitobiase and Apolipoprotein A-I displayed elevated expression levels in both the LC and LCP groups, possibly linked to lipid metabolism. Moreover, UDP-glucuronosyltransferase 2B2 showed significantly higher expression levels in the groups undergoing playpen refinement (CP and LCP), a protein which acts on various endogenous steroids. In contrast, Coronin-7 exhibited notably low expression exclusively in the Control group, indicating a unique pattern of expression within this group. Lastly, Iodotyrosine deiodinase 1 and Lon protease homolog 2 displayed a slight tendency for higher expression in the LC and LCP groups.

## Discussion

The rising population of older people globally increases the need for biomedical research on age. Rats are an established and adaptable model for many fields of science and have greatly improved the quality of medical care available [24]. However, most analyses on aged animals ignore the effects of the aging process itself, but only focus on the changes due to aging [25]. Rats are either bought aged or kept under standardized conditions until they reach the demanded age for analysis. Therefore, researchers rarely deviate from highly standardized housing conditions in fear of adding too much variability to their data [26]. Especially aging animals, which can grow substantially in size, may experience stress from inadequate cage sizes and lack of stimuli. Yet, researchers have not just a responsibility to take animal welfare into account but an obligation. In Germany, several national and institutional guidelines enforce enrichment strategies, which can be an additional financial burden for researchers. Commercial solutions for small animals exist but often come at a premium price and, therefore, disincentivize husbandry refinement [27, 28].

In our exploratory approach, we established a low-budget method to improve the husbandry conditions for aging rats based on an existing rabbit cage. Reappropriating existing material, which would be present in most animal facilities, seemed the easiest way to save money and allow for environmental enrichment. Nevertheless, the existing model has one crucial limitation: a rabbit cage would cost around €5,000 to €8,000 if this type of cage was not accessible. In our case the number of animals in a former rabbit box was related to the study design. We kept a total of 6 Sprague Dawley rats in one of the two-staged rabbit cages (**Fig 1**), but it would have been possible to group 8–10 males of that size as all animals can use the total area of the rabbit box. In total 16–20 males can be housed in such an adapted stacked rabbit rack with two boxes. On the same space of the stacked rabbit cage, 8 Type IV cages could be arranged. With our animals weighing over 700 grams at the start of the experiment, two animals per Type IV cage could be kept following the *EU Guidelines for Establishments and for the Care and Accommodation of Animals* which the commercial provider also advised [29]. Thus, utilizing rabbit cages for rat housing will not take more capacity in animal rooms as long as research projects allow to handle the animals in large groups.

Initially, cleaning the larger cage took longer than cleaning the Type IV cages due to the unavailability of a backup cage to preparate for animal transfer. Usually, the rats are just relocated into fresh Type IV cages prepared beforehand. Finally, a second cage was converted to solve this problem. The provision of prepared new cage racks in the rabbit systems increases the required storage space capacity and incurs high costs due to the additional cages required. Subsequently, the cage system was used for another group of rats after the experiment finished. The caretakers could decrease the changing interval to 10–14 days, thus achieving significant time savings.

Surveys among scientists have revealed an opinion that environmental enrichment unavoidably requires much higher costs and effort [26]. In our project, we could show that this does not have to be true. Neither did the cage modification create a big commitment of financial resources, nor was the effort in form of time and personnel requirements tremendously higher after the implementation of a system working for our facility. Using existing materials, we developed a low-cost and reversible way to provide social housing with more space and opportunities for foraging and climbing for our rats. These refinement measures enhanced the already provided enrichment in our facility, which consisted of shelters, nesting material, and gnawing opportunities. Many of these consistently suggested methods are well studied and the metareview about environmental enrichment for rats and mice in laboratories argues in favor to regarding them "as basic to good rodent housing conditions" rather than as enrichment [30]. Therefore, we aimed to describe the material used very thoroughly to allow a comprehensive understanding of our concept of environmental enrichment.

Similar to *Baumans et al.*, statements of animal care staff about increased workload after introduction of enrichment strategies such as shelters, we observed some reluctance in our facility about the added effort in cleaning and re-organizing the larger cages [31]. At the start of the experiment, a doctoral student was assigned to this task, but as the potential of this husbandry to become a new option for aging projects in rats became evident, the animal care staff took over. After two weeks of familiarization to the system, care takers reported it as not being an additional workload anymore and declared improved own satisfaction by the impression of more satisfied animals. In theory, instead of modifying a second cage to speed up the cleaning, the animals could have been put in the playground while cleaning the cage as a regular method. This way, an opportunity for positive engagement with the animals by the care takers or researchers would have been created [32], supporting the prevention of compassion fatigue and burnout in staff [33].

Regrettably, larger cages hinder animal health monitoring, especially regarding visible injuries. Dim lighting within these cages makes visual inspections and individual identification challenging. Yet, the subdued lighting aligns with rats' nocturnal nature, potentially reducing stress and enhancing their well-being compared to transparent grid-hooded cages [34, 35].

Cortisol analysis in hair samples is one established noninvasive approach for assessing stress exposure over prolonged periods rather than relying on single time points [36–38]. It has been widely used in a variety of experiments. Nevertheless, doubts have emerged regarding its suitability for making assumptions about stress exposure over several weeks to months or even years [39]. In our study, the levels of hair corticosterone, as the primary glucocorticoid hormone in rats, initially decreased, then increased, with no statistically significant differences among groups. However, group LCP, with the most refinement, exhibited the highest increase overall. We are uncertain about the exact reason for the decrease after 3 months, but we suspect it is related to a change in stress levels in our animal research facility compared to the supplier's housing conditions. While we introduced and monitored environmental changes, glucocorticoid release depends on a variety of factors such as the circadian rhythm, social hierarchy and dominance [40, 41]. It is also not suitable to discriminate distress from stimuli- and activity depriving husbandry from eustress induced cortisol excretion. The analysis of hairs with its accumulation-based results limits the informative character to conclude for distress reduction.

*Morley-Fletcher et al.* and *Mohammed et al.* reported on a decrease of corticosterone secretion through environmental enrichment after prenatal stress, similarly to our results [42, 43]. Nevertheless, despite our environmental enrichment strategies, corticosterone levels did not differ between groups at the measurement timepoints. Hair corticosterone and testosterone levels correlated over the study period, with notable correlations observed in different groups

at various time points. Testosterone levels are equally confounded by social hierarchy, age-related changes, diet, and health status [44]. The difference after 6 months of environmental enrichment was however striking, with the lowest levels in the control group. We assume social hierarchy to play an important role, as all three groups had either a larger group of other rats to contest with or were exposed to a larger group during playpen time [45, 46]. The inter-individual differences in steroid hormone levels within the groups support the hypothesis of these influences further, indicating food competition or social ranking.

Even though our handling analysis didn't find significant differences, it was subjectively observed that rats provided with a larger cage, social environment, and plenty of playpen activities were more active, curious, and easier to interact with, without any direct training. Positive interaction experiences between rats and humans can also be considered a type of enrichment, promoting increased receptivity to handling in general. Cloutier et al. reported on the benefits of tickling for rats, as signified by ultrasonic vocalizations and the open field test [47], illustrating such positive interaction.

Utilizing the liver as a representative organ to assess metabolic changes and response to our different enrichment models, we conducted qualitative and semi-quantitative proteomic analysis. Although in gene ontology analysis, we did not see any overexpressed pathways, we did find notable proteins that were overexpressed in our enriched groups relative to our control group. These were especially proteins involved in lipid metabolism such as Apolipoprotein A-I. Apolipoprotein A-I is an important part in the high-density lipoprotein (HDL) and hence, plays a role in cholesterol removal [48]. Furthermore, evidence supports its role in the inhibition of inflammatory responses [49, 50]. The lower levels of Acyl-CoA 6-desaturase in the control group may be attributed to their sedentary lifestyle compared to the enriched groups, which were more active. Furthermore, Acyl-CoA 6-desaturase exhibited a nearly linear relationship between its expression levels and the degree of refinement. In mice knockout of Acyl-CoA 6-desaturase (FADS2) has been associated with an impaired lipid metabolism [51]. *Witt et al.* demonstrated similar results in mouse and human samples and showed the connection to steroidogenesis, as FADS2 regulates polyunsaturated fatty acid content in adrenocortical cells [52].

The practice of standardizing the environment to enhance reproducibility and minimize animal use in research projects has been repeatedly criticized [53–56]. The discrepancy in defining environmental enrichment has caused researchers to compare results obtained under various husbandry conditions without adequately addressing the potential confounding effects of these differences. Consequently, the comparability of results obtained through animal experimentation has not been contested, even though different facilities use diverse external variables like food brands (high range in macronutrients), bedding, and cage systems. However, the anticipated reproducibility and applicability to other species have not met expectations, [57–59], prompting a critical re-evaluation of this construct in modern times [56]. Instead of rethinking existing research animal husbandry standards to address animal well-being in preclinical studies, the prevailing trend has been to further minimize external factors, neglecting the growing body of evidence on individual animal responses and physiological changes [60]. As a result, internal variation has even increased, as animals have a narrower range of stress regulation mechanisms available, while each individual animal has different needs. Consequently, while some animals may have their needs met, others may struggle to manage the stress, as evidenced by the wide range of behaviors exhibited by wild animals [61–63]. In aging studies, housing animals under such conditions particularly for long periods of time results in cumulative stress throughout the animal's life, creating a substantial burden even without considering interventions. Therefore, aging animals under such conditions would require strong justification, to avoid immediate rejection by the governmental authorities and welfare officers, as any suffering must be explained in terms of scientific progress.

## Limitations

This study has several limitations that must be considered. For ethical reasons related to the principles of reducing, refining, and replacing animal testing (3Rs), a small sample size was chosen, impacting the generalizability of the findings. The exploratory study design included protocol variations to avoid standardizing the environment and reduce internal variability in the animals, potentially introducing some unpredictability into the results. The animals in the study were 12 months old, which is typically considered mature, but not old, primarily due to cost and logistical constraints. These findings may not be applicable to older animals, and further research is needed to investigate the influence of these refinement methods over longer periods of time or even the whole life. Although the work group's specific interests led to a focus on liver tissue in the study, this also highlights the need for further research to investigate the effects on other organs. The results of this study may solely be applicable to Sprague Dawley rats, and further research is needed to determine their generalizability to other rat strains or small animal species. Although our study offers valuable insights into the benefits of enrichment strategies for male rats, extending these findings to female rats requires careful examination of their distinct behavioral and physiological characteristics. These include unique social dynamics and foraging behaviors. Hence, modifications in environmental enrichment might elicit gender-specific changes in social hierarchy, stress response, and activity levels. Nevertheless, the use of male rats in our study aligns with established research practices and provides relevant insights for similar male-focused studies. Future research should aim to include female rats to ensure a comprehensive understanding of enrichment's impacts across genders.

## Conclusion

Husbandry refinement for aging rats is simple and can be inexpensive, with no apparent detrimental effects on stress or development when departing from standardized husbandry. We did not find an adverse impact on our liver focused research and the variability of data was not increased by the enrichment measures. We did not find reasons to return to the former depriving housing standards represented by the control group with respect to long-term housing. Additionally, although the animals received no explicit handling training, they were subjectively calmer and could hypothetically be treated more easily.

In the light of our findings, we wholeheartedly advocate for a paradigm shift in the housing standards for aging rats in scientific research. We propose that research institutions be incentivized encouraged to integrate enrichment strategies into their animal housing protocols, simultaneously enhancing animal welfare and the reliability of scientific data. Furthermore, we encourage other researchers to replicate and expand upon our study, particularly focusing on different species, age groups, and sexes. Moreover, regular updates to animal housing standards based on new research findings are crucial to ensure that welfare practices evolve alongside scientific and ethical advancements.

## Supporting information

**S1 Table. Baseline and laboratory data of study rats.** DHEA: dehydroepiandrosterone, ALT: alanine aminotransferase, AST: aspartate aminotransferase, AP: alkaline phosphatase, LDH: lactate dehydrogenase.
(PDF)

## Acknowledgments

We want to thank our local animal welfare officer Priv.-Doz. Dr. Juliane Unger for providing her insight for the presented study. We acknowledge financial support from the Open Access Publication Fund of Charité –Universitätsmedizin Berlin.

## Author Contributions

**Conceptualization:** Dietrich Polenz, Simon Moosburner.

**Data curation:** Nathalie N. Roschke, Dietrich Polenz, Oliver Klein, Joseph M. G. V. Gassner, Simon Moosburner.

**Formal analysis:** Nathalie N. Roschke, Oliver Klein, Simon Moosburner.

**Funding acquisition:** Dietrich Polenz, Simon Moosburner.

**Investigation:** Nathalie N. Roschke.

**Methodology:** Karl H. Hillebrandt, Oliver Klein, Joseph M. G. V. Gassner, Felix Krenzien, Igor M. Sauer.

**Project administration:** Dietrich Polenz, Joseph M. G. V. Gassner, Felix Krenzien, Igor M. Sauer, Nathanael Raschzok.

**Resources:** Karl H. Hillebrandt, Dietrich Polenz, Oliver Klein, Joseph M. G. V. Gassner, Johann Pratschke, Felix Krenzien, Igor M. Sauer, Nathanael Raschzok.

**Supervision:** Johann Pratschke, Felix Krenzien, Nathanael Raschzok.

**Validation:** Karl H. Hillebrandt.

**Visualization:** Simon Moosburner.

**Writing – original draft:** Nathalie N. Roschke, Simon Moosburner.

**Writing – review & editing:** Karl H. Hillebrandt, Johann Pratschke, Felix Krenzien, Igor M. Sauer, Nathanael Raschzok, Simon Moosburner.

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
