## [Decision Letter · Decision Letter 0]

25 Dec 2023

PONE-D-23-37665Optimizing Environmental Enrichment for Sprague Dawley Rats: Exemplary Insights into the Liver ProteomePLOS ONE

Dear Dr. Moosburner,

Thank you for submitting your manuscript to PLOS ONE. After careful consideration, we feel that it has merit but does not fully meet PLOS ONE’s publication criteria as it currently stands. Therefore, we invite you to submit a revised version of the manuscript that addresses the points raised during the review process.

**ACADEMIC EDITOR:** Please do make required changes in the manuscript. 

We look forward to receiving your revised manuscript.

Kind regards,

Prof. Dr. Dragan Hrncic, MD, PhD

Academic Editor

PLOS ONE

4. We note that Figure 1B,1C and 1D in your submission contain copyrighted images. All PLOS content is published under the Creative Commons Attribution License (CC BY 4.0), which means that the manuscript, images, and Supporting Information files will be freely available online, and any third party is permitted to access, download, copy, distribute, and use these materials in any way, even commercially, with proper attribution. For more information, see our copyright guidelines: http://journals.plos.org/plosone/s/licenses-and-copyright.

a. You may seek permission from the original copyright holder of Figure 1B,1C and 1D to publish the content specifically under the CC BY 4.0 license. 

Reviewers' comments:

Reviewer's Responses to Questions

**Comments to the Author**

1. Is the manuscript technically sound, and do the data support the conclusions?

Reviewer #1: Yes

2. Has the statistical analysis been performed appropriately and rigorously? 

Reviewer #1: Yes

3. Have the authors made all data underlying the findings in their manuscript fully available?

Reviewer #1: Yes

4. Is the manuscript presented in an intelligible fashion and written in standard English?

Reviewer #1: Yes

5. Review Comments to the Author

Reviewer #1: The authors sought to establish a cost-effective method for environmental enrichment, utilizing the liver as a representative organ to assess metabolic changes in response to differing enrichment levels.

This is an interesting study with interesting observations. I would however like to invite authors to consider the following comments/suggestions

1. Perhaps, the authors may consider being robust on the recommendations and applications of the enrichment strategies in their discussion.

2. Considering that only male rats were involved, would observations vary significantly if female counterparts were involved? Would enrichment strategies require adjustments?

6. PLOS authors have the option to publish the peer review history of their article (what does this mean?). If published, this will include your full peer review and any attached files.

Reviewer #1: **Yes: **Ntethelelo Sibiya

---

## [Author Response · Author response to Decision Letter 0]

28 Dec 2023

Point-by-Point Response

Academic editor

When submitting your revision, we need you to address these additional requirements. Please ensure that your manuscript meets PLOS ONE's style requirements, including those for file naming. 

We have revised the manuscript, title page and file naming to match the journal requirements as provided in the PLOS ONE style requirements samples. 

To comply with PLOS ONE submissions requirements, in your Methods section, please provide additional information regarding the experiments involving animals and ensure you have included details on (1) methods of sacrifice, (2) methods of anesthesia and/or analgesia, and (3) efforts to alleviate suffering.

We have added the following to our methods section:

Animals were sacrificed and livers were procured under deep anesthesia with inhaled isoflurane and subcutaneous application of buprenorphine and ketamine as previously described (19-21). In short, after checking for the absence of a pain stimulus, the rat's abdomen was sterilized and opened. Blood collection and liver flushing were facilitated by cannulating the abdominal aorta, and the thoracic aorta was clamped. The liver was flushed with 20 mL of cold saline solution, both through the aorta and the portal vein. The liver was then fully isolated, removed and tissue samples snap frozen or placed in 4% formalin.

Please amend either the abstract on the online submission form (via Edit Submission) or the abstract in the manuscript so that they are identical.

We have amended the abstract in the online submission form to match the manuscript file.

We note that Figure 1B,1C and 1D in your submission contain copyrighted images. All PLOS content is published under the Creative Commons Attribution License (CC BY 4.0), which means that the manuscript, images, and Supporting Information files will be freely available online, and any third party is permitted to access, download, copy, distribute, and use these materials in any way, even commercially, with proper attribution. 

The images in figures 1B, 1C, and 1D are our own and were taken by the first author of the manuscript Ms. Nathalie Roschke. We have added a paragraph in the figure legend to reflect this. 

[…] Images by Nathalie N. Roschke [2023]

Please review your reference list to ensure that it is complete and correct. If you have cited papers that have been retracted, please include the rationale for doing so in the manuscript text or remove these references and replace them with relevant current references. Any changes to the reference list should be mentioned in the rebuttal letter that accompanies your revised manuscript. If you need to cite a retracted article, indicate the article’s retracted status in the References list and also include a citation and full reference for the retraction notice.

We have reviewed the reference list and updated two ISBN numbers for books we cited. We are not aware of any retracted articles. These are the changed references:

Kaliste E. Animal Welfare, The Welfare of Laboratory Animals. 2007. ISBN 978-1-4020-2270-8.

United Nations, Department of Economic and Social Affairs | Population Division. World population ageing. 2015. ISBN 978-92-1-057854.

Reviewer

The authors sought to establish a cost-effective method for environmental enrichment, utilizing the liver as a representative organ to assess metabolic changes in response to differing enrichment levels. This is an interesting study with interesting observations. I would however like to invite authors to consider the following comments/suggestions: 

Perhaps, the authors may consider being robust on the recommendations and applications of the enrichment strategies in their discussion. 

Thank you for your insightful comments on our manuscript, we hope to have addressed your concerns with the following paragraph we have added to our discussion section:

In light of our findings, we strongly advocate for a paradigm shift in the housing standards for aging rats in scientific research. We recommend that research institutions are encouraged to integrate enrichment strategies in their animal housing protocols, enhancing both animal welfare and the reliability of scientific data. Furthermore, we encourage other researchers to replicate and expand upon our study, particularly focusing on different species, age groups, and sexes. Additionally, regular updates to animal housing standards based on new research findings should ensure that welfare practices evolve alongside scientific and ethical advancements.

Considering that only male rats were involved, would observations vary significantly if female counterparts were involved? Would enrichment strategies require adjustments?

The inclusion of only male rats in our study does indeed raise important questions about the applicability of our findings to female rats, as well as the potential need for adjustments in enrichment strategies to accommodate gender-specific behaviors and needs. While there are behavioral and physiological differences between male and female rats, the enrichment strategies employed in our study are likely to be beneficial for both genders. These strategies, such as improved housing conditions and opportunities for social interaction and physical activity, are universally advantageous for rodent welfare. Nevertheless, our study exclusively involved male rats, as they are the most common choice for research settings due to their well-documented characteristics. This provides a clearer understanding of the enrichment impact in a male-only context, which is valuable given their frequent use in research. To reflect these aspects and the potential need for future research to explore gender-specific responses to environmental enrichment, we have added the following context in the discussion section of our manuscript:

Additionally, while our study offers valuable insights into the benefits of enrichment strategies for male rats, extending these findings to female rats requires careful consideration of their unique behavioral and physiological characteristics. These include unique social dynamics, and foraging behavior. Hence changes in environmental enrichment might display gender-specific changes in social hierarchy, stress response, and activity levels. Nevertheless, the use of male rats in our study is consistent with common research practices and provides relevant insights for similar male-focused studies. Future research should aim to include female rats to ensure a comprehensive understanding of enrichment impacts across genders.

---

## [Decision Letter · Decision Letter 1]

8 Jan 2024

Optimizing Environmental Enrichment for Sprague Dawley Rats: Exemplary Insights into the Liver Proteome

PONE-D-23-37665R1

Dear Dr. Moosburner,

We’re pleased to inform you that your manuscript has been judged scientifically suitable for publication and will be formally accepted for publication once it meets all outstanding technical requirements.

Kind regards,

Prof. Dr. Dragan Hrncic, MD, PhD

Academic Editor

PLOS ONE

Additional Editor Comments (optional):

Reviewers' comments:

Reviewer's Responses to Questions

**Comments to the Author**

1. If the authors have adequately addressed your comments raised in a previous round of review and you feel that this manuscript is now acceptable for publication, you may indicate that here to bypass the “Comments to the Author” section, enter your conflict of interest statement in the “Confidential to Editor” section, and submit your "Accept" recommendation.

Reviewer #1: All comments have been addressed

2. Is the manuscript technically sound, and do the data support the conclusions?

Reviewer #1: Yes

3. Has the statistical analysis been performed appropriately and rigorously? 

Reviewer #1: Yes

4. Have the authors made all data underlying the findings in their manuscript fully available?

Reviewer #1: Yes

5. Is the manuscript presented in an intelligible fashion and written in standard English?

Reviewer #1: Yes

6. Review Comments to the Author

Reviewer #1: (No Response)

7. PLOS authors have the option to publish the peer review history of their article (what does this mean?). If published, this will include your full peer review and any attached files.

Reviewer #1: **Yes: **Ntethelelo Sibiya

---

## [Editor Report · Acceptance letter]

6 Feb 2024

PONE-D-23-37665R1 

PLOS ONE

Dear Dr. Moosburner, 

I'm pleased to inform you that your manuscript has been deemed suitable for publication in PLOS ONE. Congratulations! Your manuscript is now being handed over to our production team.

Kind regards, 

on behalf of

Professor Dragan Hrncic 

Academic Editor

PLOS ONE